# GENERATIVE HINTS

## ABSTRACT

Data augmentation is widely used in vision to introduce variation and mitigate overfitting, through enabling models to learn invariant properties, such as spatial invariance. However, these properties are not fully captured by data augmentation alone, since it attempts to learn the property on transformations of the training data only. We propose *generative hints*, a training methodology that directly enforces known invariances in the entire input space. Our approach leverages a generative model trained on the training set to approximate the input distribution and generate *unlabeled* images, which we refer to as *virtual examples*. These virtual examples are used to enforce functional properties known as *hints*. In generative hints, although the training dataset is fully labeled, the model is trained in a semi-supervised manner on both the classification and hint objectives, using the *unlabeled* virtual examples to guide the model in learning the desired hint. Across datasets, architectures, and loss functions, generative hints consistently outperform standard data augmentation when learning the same property. On popular fine-grained visual classification benchmarks, we achieved up to 1.78 % top-1 accuracy improvement (0.63% on average) over fine-tuned models with data augmentation and an average performance boost of 1.286 % on the CheXpert X-ray dataset.

## 1 INTRODUCTION

Data augmentation was first introduced in LeCun et al. (1989), using translations, scalings, and rotations to improve recognition robustness. This established a precedent for its widespread use in vision classification (Perez & Wang, 2017; Shorten & Khoshgoftaar, 2019). In practice, data augmentation applies transformations such as color jitter, rotation, or translation to an image, and the augmented sample is trained with the same label as the original. The model attempts to learn the corresponding invariances from these augmentations, but fails to fully capture the relationships. This occurs to varying degrees in different models. It has been attempted to build them into architectures as an inductive bias. For example, convolutional neural networks (CNNs) He et al. (2016) exhibit an inductive bias toward spatial invariance through convolutions. Building on this, transformer-based architectures (Vaswani et al., 2017) particularly benefit from data augmentation because they lack such built-in inductive biases. For example, the Vision Transformer (ViT) (Dosovitskiy et al., 2021) encodes images as patches, which enables efficient representation learning but does not inherently enforce spatial invariance. To address this, the Swin Transformer (Liu et al., 2021) introduced a hierarchical design inspired by CNNs, partially reintroducing spatial invariance into transformer-based vision models.

Nonetheless, these architecture changes and data augmentation are not enough to learn the corresponding properties. We offer a solution that directly enforces the property through *jointly training* with the classification objective in fully labeled data settings. Specifically, we introduce a methodology called *generative hints*.

Following the definition in Abu-Mostafa (1990), a hint is any known property known for the target function we are modeling. Originally, hints were applied to tabular data, using random noise to enforce the property. In vision, however, images are high-dimensional, and random noise lies far from the true input distribution, making this strategy ineffective. To address this, we approximate the input distribution through training a generative model on the training dataset.

We sample *unlabeled* images from the generative model, *virtual examples*, and apply the hint function to them. This enables us to generate unlimited examples from the input distribution without being restricted to the finite training set. We train the model in a semi-supervised manner on both the classification and hint objectives, guiding the learning of the hint property through *virtual examples*. Using our generative hint methodology, we consistently outperform standard data augmentation while explicitly teaching the model the intended invariance. Specifically, we make the following contributions:

1. By applying generative hints, existing models across architectures, datasets, and objective functions consistently *outperforms* standard data augmentation. *Generative hints* achieves up to 1.78% accuracy (0.63% on average) and 1.28% on average over standard data augmentation on finegrain visual classification and CheXpert, respectively.

2. To our knowledge, we are the first to reformulate a fully supervised classification task into a semi-supervised learning task in fully labeled training sets by treating data synthesized from a generative model as *unlabeled data*.

3. Our method introduces a way to jointly learn known properties of the target function directly learning them over the input entire input space.

## 2 RELATED WORK

### 2.1 GENERATIVE MODELS FOR VISION

**Generative Models** Recent advances in generative modeling have enabled the synthesis of high-fidelity images from noise, primarily through diffusion models (Ho et al., 2020; Song et al., 2021; Rombach et al., 2022) and GANs (Goodfellow et al., 2014; Karras et al., 2019; 2020b; 2021). These models have been applied both as tools for data generation and as components of downstream training pipelines. In classification, discriminators have been adapted for semi-supervised learning (Kingma et al., 2014; Radford et al., 2016), while synthetic data has been used to expand training sets in medical and natural image domains (Antoniou et al., 2017; Frid-Adar et al., 2018b; Zhao et al., 2019; Azizi et al., 2023; Yuan et al., 2024). More recently, diffusion-based pipelines (Bordes et al., 2023; Huang et al., 2023; Zhang et al., 2024) highlight the ability of generative models to provide controllable, task-aware augmentation. However, these approaches primarily focus on increasing data diversity rather than directly enforcing functional properties.

**Data Augmentation and Invariances** Conventional data augmentation is widely used to induce invariances (e.g., spatial or color invariance) by perturbing training examples. While effective for regularization, this strategy only encourages models to learn invariance indirectly, relying on the hope that augmented samples approximate invariance-preserving transformations (Perez & Wang, 2017; Shorten & Khoshgoftaar, 2019).

**Generative Data Augmentation** Generative data augmentation (GDA) builds on generative models to synthesize additional labeled data, with demonstrated benefits in low-data regimes, domain-specific applications, and joint generation–classification frameworks (Mahapatra & Ge, 2022). Yet, existing GDA methods treat generated examples primarily as extra training data, without using them to explicitly encode known invariances or functional constraints.

Unlike standard augmentation or GDA, generative hints use synthetic examples as unlabeled carriers of functional properties. That is, generative hints focus is on learning properties of the target function through our semi-supervised training so it can be additively done with existing GDA works.

### 2.2 HINTS

*Hints* were first introduced by Abu-Mostafa (1990; 1995) to teach machine learning models functional properties of the target function and data. These properties, referred to as hints, are incorporated as auxiliary objectives optimized alongside the main task. For example, in credit default prediction using tabular data, the target is to predict whether a default will occur given input features. Domain knowledge provides that, if all other features remain fixed while income increases, the probability of default should decrease. This property can be formalized as a *monotonicity hint* and enforced through an auxiliary loss. Similarly, in the foreign exchange (FX) markets, a *symmetry*

*hint* has been used to regularize models against noisy financial data, leading to significantly improved annualized returns. Generative Hints is explicitly different from previous iterations of hints in its formulation of using a generative model to represent the input space to learn the functional properties.

## 3    WHAT ARE HINTS?

### 3.1    PROBLEM STATEMENT

We begin by defining $f$, $X$, $Y$, $D_{train}$, and $D_{test}$ as the true underlying function, input distribution, output distribution, training set, and the test set, respectively. In the case of image classification, $X$ corresponds to the distribution of images and $Y$ to the class probability distribution. While we focus on vision tasks, these definitions naturally extend to other modalities and problem settings. In Definitions 1 and 2, we formally introduce the general notion of a hint, as well as the specific case of an invariance hint.

**Definition 1 (Hint)** *A hint is a known property of the target function $f$ expressed through a transformation of the input. Formally, let $h$ be a hint function such that $h(x) = x'$. A hint specifies a known relationship between $f(x)$ and $f(x')$, which can be enforced during training as an auxiliary objective.*

**Definition 2 (Invariance Hint)** *An invariance hint specifies that the output of the target function $f$ remains unchanged under a transformation of the input. Formally, for a hint function $h$ and any $x \in X$, let $h(x) = x'$. Then the invariance hint enforces that $f(x) = f(x')$.*

While both data augmentation and generative hints aim to teach a model invariance, their mechanisms are fundamentally different. Data augmentation implicitly teaches invariance by applying transformations to labeled training examples. Our method, in contrast, explicitly enforces a functional property on unlabeled virtual examples via an auxiliary objective.

### 3.2    ENFORCING HINTS THROUGH VIRTUAL EXAMPLES

Applying hints directly on training data can lead to overfitting, where the model memorizes the hints with respect to specific training examples rather than learning the underlying property. Moreover, this approach conflates supervised learning on the labels with hint-based learning of the functional property. To address this, we apply hints to *virtual examples* instead.

A virtual example serves as an input to which a hint is applied, analogous to how a training example is input to an objective function. To ensure virtual examples are representative, we sample unlabeled images from a generative model trained on the input distribution. This is fundamentally different from the previous definition which used only random noise in tabular settings. Formally, we define a virtual example in Definition 3 and an invariance hint applied to virtual examples in Definition 4.

**Definition 3 (Virtual Example)** *A virtual example $x_v$ is an unlabeled input generated by a generative model $G$ trained on the training set $D_{train}$.*

**Definition 4 (Invariance Hint on Virtual Examples)** *Given a generative model $G$, an invariance hint is defined via a hint transformation function $h$ such that $h(x_v) = x_v'$ and the target function satisfies $f(x_v) \approx f(x_v')$ for $x_v \in G$.*

Leveraging the known ability of generative models to approximate the input distribution, we sample virtual examples from a generative model trained on the input distribution, thereby adapting the original hint methodology to the high-dimensional image domain.

Specifically, we employ two types of invariance hints: a flip-invariant hint and a spatial-invariant hint, defined formally in Definitions 5 and 6, respectively. These invariances are commonly used in data augmentation to create duplicate training examples, and they correspond to properties that image classification functions should naturally respect; that is, the predicted class distribution is expected to remain unchanged under these transformations.

**Definition 5 (Flip Invariance Hint)** *Let $h$ be a function that horizontally flips an image. A flip invariance hint asserts that, for any virtual example $x_v \in G$, the target function satisfies*

$$f(h(x_v)) = f(x_v).$$

**Definition 6 (Spatial Invariance Hint)** *Let $h$ be a function that translates and rotates an image by factors $a_t$ and $a_r$, respectively. A spatial invariance hint asserts that, for any virtual example $x_v \in G$ and for $(a_t, a_r) \in A$, where $A$ is the set of small, non-aggressive spatial transformations, the target function satisfies*

$$f(h(x_v)) = f(x_v).$$

## 4 GENERATIVE HINTS ALGORITHM

### 4.1 TRAINING GENERATIVE MODELS EFFICIENTLY

We use StyleGAN3 from Karras et al. (2021) as our generative model to produce virtual examples. This model generates unlabeled images from the input distribution without class conditioning. We chose StyleGAN3 due to its strong performance across image generation tasks and dataset sizes. While other generative models, including diffusion models, could be used, StyleGAN3 provides a favorable balance between sampling efficiency and image quality.

Training generative models in limited data settings requires careful data-efficient strategies to prevent overfitting. We leverage adaptive discriminator augmentation (ADA) from Karras et al. (2020a), which adjusts the strength of data augmentations dynamically based on overfitting signals, improving image quality in low-data regimes.

We extend ADA with a curriculum learning approach. Let $A_s, A_w, A_n$ denote strong, weak, and no augmentations, respectively. Training proceeds sequentially: starting with $A_s$ to provide a larger, more diverse distribution for initial learning, followed by $A_w$ and finally $A_n$. At each stage, the augmentation strength is decreased after convergence. The augmentations used include: `xflip`, `rotate90`, `xint`, `scale`, `rotate`, `anisco`, `xfrac`, `brightness`, `contrast`, `lumaflip`, `hue`, and `saturation`, with each set ($A_s$, $A_w$, $A_n$) being a subset of these operations. We found this setup to allow for consistently strong performance for image generation across datasets.

### 4.2 HINT LOSS FUNCTION

To enforce invariance hints, we measure the similarity between the model's predictions on the original and transformed inputs using the symmetric Kullback-Leibler (KL) divergence. The symmetry ensures that both the original and hint-adjusted distributions are treated equally. Moreover, using a KL-based loss aligns well with the cross-entropy loss, since the gradient of cross-entropy with a one-hot target is equivalent to KL divergence. We introduce a temperature parameter to more strictly enforce alignment between the distributions. The formal definition of the virtual symmetric KL divergence loss is given below.

**Definition 7 (Symmetric KL Hint Loss)** *Let $h$ denote a hint transformation applied to a virtual example $x_v \in G$, producing $x'_v = h(x_v)$. Let the model's predicted probability distributions be $\hat{f}(x_v) = p$ and $\hat{f}(x'_v) = q$, where $\hat{f}$ is the model under training. The hint loss using symmetric KL divergence is defined as:*

$$\mathcal{L}_{hint\text{-}ce}(p, q) = \frac{1}{2}\Big(\text{KL}\Big(\frac{p}{T} \,\Big\|\, \frac{q}{T}\Big) + \text{KL}\Big(\frac{q}{T} \,\Big\|\, \frac{p}{T}\Big)\Big),$$

*where $T$ is a temperature parameter controlling the sharpness of the predictive distributions.*

In addition to the symmetric KL loss, we introduce a mean squared error (MSE) based hint loss, formalized in Definition 8. This loss provides an alternative mechanism to align model outputs under the hint transformation. It serves as an auxiliary loss alongside the main training objective, when the main objective is itself an MSE, and can also complement other objectives.

**Definition 8 (MSE Hint Loss)** *Let $h$ denote a hint transformation applied to a virtual example $x_v \in G$, producing $x'_v = h(x_v)$. Let the model's predicted logits be $\hat{f}(x_v) = y_v$ and $\hat{f}(x'_v) = y'_v$, where $\hat{f}$ is the model under training. The MSE-based hint loss is defined as:*

$$\mathcal{L}_{hint\text{-}mse}(y_v, y'_v) = \frac{1}{d} \sum_{i=1}^{d} \left( y_{v,i} - y'_{v,i} \right)^2,$$

*where $d$ is the dimensionality of the output logits.*

### 4.3 TRAINING ALGORITHM

Our approach follows a multi-objective learning framework, optimizing both classification objective via cross-entropy loss and the hint objective. Labeled training data from $D_{train}$ is used for the classification loss, while unlabeled images sampled from the generative model $G$ serve as virtual example inputs for the hint objective. Optimization alternates between the two objectives, with each batch switching evenly between the classification and hint losses. The full procedure is summarized in Algorithm 1. Notably, the virtual examples from $G$ are generated on-the-fly from Gaussian noise for each batch, rather than precomputed, ensuring diversity and reducing memory requirements.

---

**Algorithm 1** Generative Hints Training Algorithm

---

Training set $\mathcal{D}_{\text{train}} = \{(x_i, y_i)\}_{i=1}^{N}$
Classifier $\hat{f}$, hint transformation $h$, generative model $G$
Classification loss $L_{\text{class}}$, hint loss $L_{\text{hint}}$, coefficient $\alpha$
Number of epochs $E$
**for** epoch $e = 1, \ldots, E$ **do**
    **for** mini-batch $b \subset \mathcal{D}_{\text{train}}$ **do**
        Update $\hat{f}$ on $b$ using $L_{\text{class}}$
        Sample virtual example $x_v \sim G$
        $x'_v \leftarrow h(x_v)$
        $y_v \leftarrow \hat{f}(x_v), \quad y'_v \leftarrow \hat{f}(x'_v)$
        Update $\hat{f}$ using $\alpha \cdot L_{\text{hint}}(y_v, y'_v)$
    **end for**
**end for**

---

We introduce a coefficient $\alpha$ to scale the hint loss, controlling its relative weight compared to the classification objective. This weighting is necessary because the gradients and learning dynamics of the two objectives can differ significantly. In our experiments, we found that a fixed $\alpha$ already provides stable and consistent improvements across datasets and architectures. While adaptive scheduling of $\alpha$ is a promising direction for further optimization, we show that even the simple fixed version is sufficient to validate the effectiveness of generative hints.

## 5 EXPERIMENTS AND RESULTS

We evaluated our method on four popular fine-grained visual classification datasets: Stanford Cars (Krause et al., 2013), CUB-200-2011 (Caltech Birds) (Wah et al., 2011), FGVC Aircraft (Maji et al., 2013), and Oxford Flowers (Nilsback & Zisserman, 2008). Experiments were conducted using two model architectures: ViT-B (Dosovitskiy et al., 2021; Vaswani et al., 2017) and Swin-B (Liu et al., 2021), chosen for their strong performance on fine-grained classification and to demonstrate the generality of our approach across architectures.

We further evaluated generative hints in a medical imaging setting using the CheXpert dataset (Irvin et al., 2019) with a ResNet50 (He et al., 2016), employing mean squared error as the training objective. Finally, we performed an ablation study to examine the impact of generative model quality on classification performance. All experiments were conducted on a single NVIDIA H100 GPU, training both the generative and classification models.

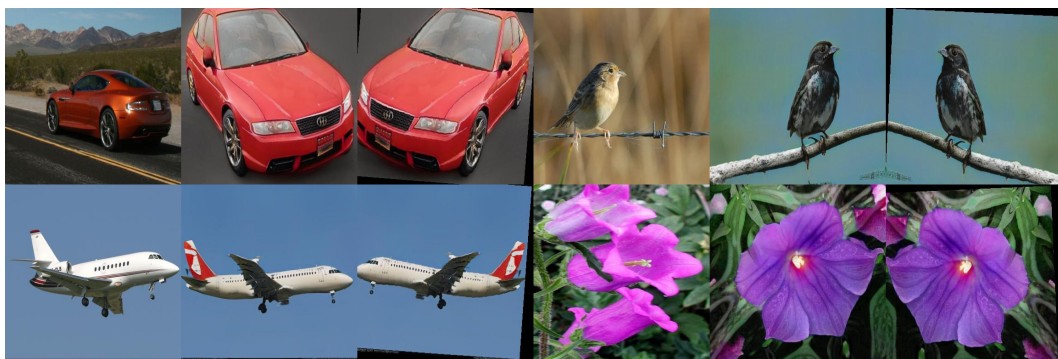

Figure 1: Depiction of virtual examples applied to each dataset. The datasets shown are Stanford Cars (top left), CUB-200-2011 Caltech Birds (top right), FGVC Aircraft (bottom left), and Oxford Flowers (bottom right). For each dataset, from left to right, we show an original training image, a virtual example sampled from the generative model, and the corresponding hint-transformed image.

## 5.1 GAN TRAINING SPECS

We used StyleGAN3 as our generative model, training a separate model on the training set of each dataset. Training followed the curriculum learning strategy described in Section 4.1, combined with adaptive discriminator augmentation (ADA). As StyleGAN3 requires resolutions that are powers of two, we trained all models at $512 \times 512$ resolution and resized images to the target model resolution to avoid information loss. Training hyperparameters included a batch size of 16, a generator learning rate of 0.0025, a discriminator learning rate of 0.001, and a gamma of 4.0. Table 1 summarizes the statistics of each dataset and the quality of the trained generative models, measured using the Fréchet Inception Distance (FID) (Heusel et al., 2017). FID quantifies the similarity between the distribution of real and generated images in the feature space of an Inception-V3 classifier. We selected the generative model with the best FID for each dataset, which was then frozen and used solely for sampling virtual examples for hint training.

Table 1: Dataset statistics for the four fine-grained visual classification benchmarks. FID is measured for StyleGAN3 trained on each training set, used for virtual example generation. Number of classes, number of training images, and total number of images in the dataset are provided as well.

| Dataset | Classes | Training Size | Total Size | FID |
|---|---|---|---|---|
| Stanford Cars | 196 | 8,144 | 16,185 | 5.29 |
| FGVC Aircraft | 100 | 6,800 | 10,200 | 4.73 |
| Caltech Birds | 200 | 5,994 | 11,788 | 7.04 |
| Oxford Flowers | 102 | 2,040 | 8,189 | 12.62 |

## 5.2 FINE-GRAIN VISION CLASSIFICATION TRAINING RESULTS

Most prior applications of generative models in vision classification either use them for data augmentation (Antoniou et al., 2018; Frid-Adar et al., 2018a) or train the generative model to perform classification directly (Azizi et al., 2023). In contrast, we use the generative model solely to approximate the input distribution. Consequently, our baseline is the best-performing result obtained using standard supervised learning with data augmentation.

We evaluated our approach using the ViT-B and Swin-B vision transformer models with patch sizes of 16 and 4, respectively. Both models were pretrained on ImageNet-1k. All experiments were conducted at a resolution of $384 \times 384$ with a batch size of 32 for both training and virtual examples. We used the AdamW optimizer with a learning rate of 0.0001 and a momentum of 0.01, training for 200 epochs with a cosine annealing learning rate scheduler. Standard data augmentations included random horizontal flipping (applied with 50% probability), translation, and rotation, with translation

and rotation factors uniformly sampled from 0–5%. This setup was chosen to maximize the baseline performance without using hints.

For the generative hints training, we sample virtual examples from the StyleGAN3 generative model in section 5.1 to enforce the hint property. We used a temperature $T = 0.8$ and performed a sweep over the fixed hint loss coefficient $\alpha = \{0.1, 0.5, 1, 5, 10, 25, 50\}$ to account for differences in gradient magnitudes between the classification and hint objectives. Hints were applied using the same training setup as the baseline, with the same transformations as data augmentation, except that horizontal flipping was applied with 100% probability to enforce the flip-invariance hint. This design allows a direct comparison between baseline augmentation and our hint-based training. Results across various models are reported in Table 2. Experiments were run for 5 seeds with the average result reported.

| Dataset | ViT-B Baseline | | ViT-B w/ Hints | | Swin-B Baseline | | Swin-B w/ Hints | |
|---|---|---|---|---|---|---|---|---|
| | Acc. | Hint L. | Acc. | Hint L. | Acc. | Hint L. | Acc. | Hint L. |
| Stanford Cars | 90.90 | 0.714 | **91.58** | **4.4e-05** | 92.92 | 0.749 | **93.53** | **2.3e-07** |
| FGVC Aircrafts | 86.43 | 0.722 | **88.21** | **1.8e-07** | 92.55 | 0.772 | **92.83** | **1.9e-07** |
| Caltech Birds | 88.45 | 0.571 | **88.76** | **4.4e-05** | 90.28 | 0.460 | **91.11** | **2.9e-07** |
| Oxford Flowers | 98.94 | 0.196 | **99.43** | **8.5e-04** | 99.61 | 0.176 | **99.68** | **3.8e-06** |

Table 2: Top-1 accuracy (Acc.) and hint loss (Hint L.) on virtual examples for the Stanford Cars, FGVC Aircraft, Caltech Birds, and Oxford Flowers datasets. Hint loss is measured using symmetric KL divergence with a temperature of $T = 1$ and is denoted as *Hint L*. Bold indicates the best performance for each model and dataset.

We observe consistent improvements across all datasets through the use of generative hints, with performance gains evident for both ViT-B and Swin-B. Table 2 also reports the hint loss computed on virtual examples using the symmetric KL divergence defined in Definition 7 with a temperature of $T = 1$. While data augmentations alone show limited performance on the generated examples, training with the hint objective substantially improves alignment and overall performance.

The hint objective acts as an additional regularizer, providing self-supervised training on virtual examples through an auxiliary task, which ensures better alignment of model predictions throughout training.

## 5.3 CHEXPERT TRAINING RESULTS

To evaluate the robustness of our algorithm, we extended our experiments to a different domain and objective function by using the CheXpert dataset (Irvin et al., 2019). CheXpert is a large-scale chest radiograph dataset collected from Stanford Hospital, containing 224,316 X-rays from 65,240 patients, annotated for 14 common thoracic pathologies as well as a "No Finding" category. For our experiments, we used 9 categories: No Finding, Enlarged Cardiomediastinum, Cardiomegaly, Lung Opacity, Pneumonia, Pleural Effusion, Pleural Other, Fracture, and Support Devices. Labels are automatically extracted from radiology reports using a rule-based NLP system, which assigns each observation as positive (1), negative (-1), or uncertain (0).

We trained a StyleGAN3 generative model on the full dataset at $256 \times 256$ resolution, achieving a FID of 4.38. We used $256 \times 256$ rather than $512 \times 512$ as done previously due to the model utilized taking in $256 \times 256$. Training used a batch size of 16, a generator learning rate of 0.0025, a discriminator learning rate of 0.001, and $\gamma = 8.0$. Adaptive discriminator augmentation was not applied, as the dataset size was large and many standard augmentations are inappropriate for X-ray images.

For classification, we used a ResNet50, differing from the previously used transformer-based models. Both the classification and hint objectives (Definition 8) were optimized using MSE loss. Input images were $256 \times 256$ grayscale, normalized with ImageNet statistics, and the model used ImageNet-pretrained weights (Deng et al., 2009). Training used a batch size of 64 with the Adam optimizer ($lr = 0.00001$, $\beta = (0.9, 0.999)$) for 5 epochs with a cosine annealing learning rate schedule. Data augmentation included translation and rotation, with factors uniformly sampled

from 0–5%. This setup optimized the baseline performance without hints. When applying generative hints, we used the same transformations as the hint with $\alpha = 0.1$. Full results are reported in Table 3 where experiments were run for 5 seeds and the average is reported.

Table 3: Classification MSE loss across multiple pathologies on the CheXpert dataset, with and without generative hints. *Percent Gain* represents the relative reduction in classification MSE from the baseline to the model trained with hints.

| Pathology | Baseline | w/ Hints | % Gain |
|---|---|---|---|
| No Finding | **0.636** | 0.639 | -0.472% |
| Enlarged Cardiomediastinum | 0.719 | **0.704** | 2.086% |
| Cardiomegaly | 0.339 | **0.337** | 0.590% |
| Lung Opacity | 0.795 | **0.784** | 1.384% |
| Pneumonia | 0.797 | **0.781** | 2.008% |
| Pleural Effusion | **0.423** | 0.425 | -0.473% |
| Pleural Other | 0.876 | **0.864** | 1.370% |
| Fracture | 0.673 | **0.660** | 1.932% |
| Support Devices | 0.983 | **0.952** | 3.154% |

Table 3 shows that on CheXpert, generative hints consistently improve performance across all pathologies, with an average improvement of **1.286 % gain**. Furthermore, even under this different domain and objective function, generative hints outperform traditional data augmentation supervised learning.

## 5.4 GENERATIVE MODEL QUALITY STUDY

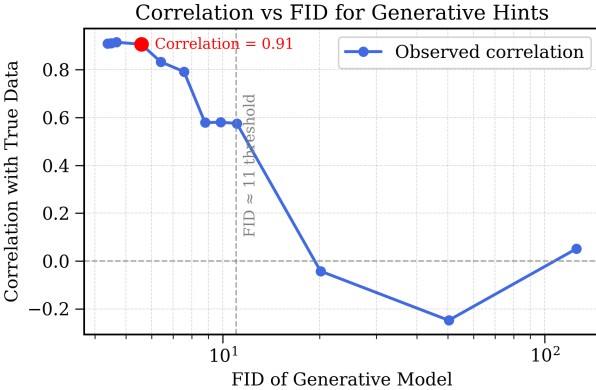

Figure 2: Correlation between the generative hint loss on generated samples and the hint loss on real training data, plotted against the FID of the generative model. The horizontal dashed line indicates zero correlation. The vertical dashed line highlights the approximate FID threshold ($\sim 11$) where the generative model begins to provide meaningful learning signal. The red point marks the FID 5.58 where correlation reaches 0.91.

We conducted an experiment to evaluate the quality of the generative model required for effectively learning the hint with respect to the training data. Specifically, we sought to determine the FID threshold at which the generative model sufficiently captures the input distribution so that the hint learned on virtual examples transfers to the real training data.

To do this, we trained models using generative hints across generative models with varying FID scores and computed the correlation between the hint loss on virtual examples and the hint loss applied to real training data. Models were trained only on virtual examples (without data augmentation

on the real examples), but we assessed the hint with respect to both virtual and real data to measure how well learning from the generative model reflects the true data distribution.

The experimental setup followed the CheXpert specification from Section 5.2, varying only the FID of the generative model. Figure 2 shows the correlation versus FID. At FID values above 50, correlation is very poor, indicating that low-quality generative models provide little value for learning the hint. Once the FID drops below 11, the correlation becomes significant, reaching 0.91 at an FID of 5.58, with no substantial gains observed for lower FIDs. These results indicate that a sufficiently high-quality generative model is necessary for effective hint learning, although moderate-quality models still provide meaningful benefits.

## 6 FUTURE WORKS

There are several promising directions for future work to expand upon the benefits of generative hints. First, we currently use a fixed scheduler to balance the weights between the classification and hint objectives, but a dynamic scheduler could potentially improve classification performance. By adapting the relative weight based on the learning rates or gradient magnitudes of the two objectives, a dynamic scheduler could better balance training and further enhance the downstream classification performance.

Second, while we focused on using hints that mirror standard data augmentation to demonstrate that the same property can be learned more effectively, hints could be designed to capture other properties of the target function that are difficult to encode via augmentation. In particular, the use of a generative model enables *embedding hints*, where noise is added directly to the latent embedding space to create augmented representations. These embeddings can be perturbed either globally or selectively along specific dimensions to generate meaningful variations in the input, as explored in Härkönen et al. (2020).

Finally, our semi-supervised framework in fully labeled datasets allows for the opportunity to be applied in different vision settings. That is, it has the potential to be applied to object detection and segmentation where it can enforce spatial invariance on the bounding box/segmentation mask.

## 7 CONCLUSION

We proposed a method to reformulate supervised classification on fully labeled datasets as a semi-supervised learning problem by treating data synthesized from a generative model, trained solely on the labeled training set, as *unlabeled data*. This approach enables models to learn functional properties, or *hints*, of the target function by applying them to virtual examples sampled from the generative model.

We evaluated our method across fine-grained visual classification and medical imaging domains, considering multiple model architectures and objective functions. Generative hints consistently *outperformed* traditional data augmentation when learning the same property explicitly assumed by the augmentation without overfitting. Moreover, we demonstrated that a perfect generative models is not required for generative hints to learn the property. We showed Generative Hints as a new and versatile tool for injecting domain knowledge into deep learning models, opening up a new avenue for research in explicit regularization and semi-supervised learning for fully labeled data settings.

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

# A APPENDIX

## A.1 DATASETS

We ran on the 4 datasets Stanford Cars, FGVC Aircrafts, Caltech Birds, and Oxford Flowers. The datasets are all fine grain visual classification datasets with 100 or more classes. Full dataset specifications can be seen in Table 2. For datasets with a training/val/test split we combined the training and validation set, which is considered the standard for the datasets.

Table 4: Summary statistics for fine-grained visual classification datasets: number of classes, total image count, and standard train/test splits.

| Dataset | # Classes | Total Images | Train Images | Test Images |
|---|---|---|---|---|
| Stanford Cars | 196 | 16,185 | 8,144 | 8,041 |
| FGVC Aircrafts | 100 | 10,200 | 6,800 | 3,400 |
| Caltech Birds (CUB-200-2011) | 200 | 11,788 | 5,994 | 5,794 |
| Oxford Flowers 102 | 102 | 8,189 | 2,040 | 6,149 |

## A.2 GENERATIVE MODEL TRAINING

Generative models were trained according to the specifications listed in Table 5, using only the training split of each dataset. All models were trained until convergence with Adaptive Discriminator Augmentation (ADA) Karras et al. (2020a), in which augmentations are applied to images before being passed to the discriminator. Training was performed on a single NVIDIA H100 GPU and continued until Fréchet Inception Distance (FID) convergence Heusel et al. (2017). The augmentations used included: `xflip`, `rotate90`, `xint`, `scale`, `rotate`, `anisco`, `xfrac`, `brightness`, `contrast`, `lumaflip`, `hue`, and `saturation`. Full augmentation specifications are available in the official StyleGAN3 repository. The resulting generative model FIDs can be observed in Table 6.

## A.3 CLASSIFICATION AND HINTS TRAINING

Both the Swin-B and ViT-B/16 transformer models were pretrained on ImageNet **?**. We trained these models using the hyperparameters reported in Table 5, which were tuned to maximize performance prior to introducing our generative hints methodology. We found for training without hints the best data augmentation combination to flip ($p = 0.5$), rotation sampled uniformly from [0, 5%], and translation sampled uniformly from [0, 5%]. When applying generative hints we applied we used the same transformation except flip ($p = 1.0$). For fairness, all learning parameters were kept fixed when applying generative hints. The only modifications were the weighting factor $\alpha$ applied to the hint loss, and the temperature $T$ used to scale the distributions in the symmetric KL loss, which was set to $T = 0.8$. Training was performed on a single NVIDIA H100 GPU.

Table 5: Training hyperparameters for the generative model (StyleGAN3 with ADA).

| Hyperparameter | Value |
|---|---|
| Model Type | StyleGAN3 |
| Resolution | $512 \times 512$ |
| Adaptive Discriminator Augmentation | Enabled |
| Mirror | Enabled |
| Optimizer | AdamW |
| Generator Learning Rate | 0.0025 |
| Discriminator Learning Rate | 0.001 |
| Batch Size | 16 |
| Gamma | 4.0 |
| Stop Condition | FID convergence |

Table 6: The resulting StyleGAN3 models trained on each of the datasets including the FID achieved.

| Dataset | # Classes | Train Images | FID |
|---|---|---|---|
| Stanford Cars | 196 | 8,144 | 4.27 |
| FGVC Aircrafts | 100 | 6,800 | 4.72 |
| Caltech Birds (CUB-200-2011) | 200 | 5,994 | 7.37 |
| Oxford Flowers 102 | 102 | 2,040 | 12.62 |

Hints were trained using a symmetric KL divergence loss as defined in Definition 2, chosen for its ability to align distributions and its close relationship to cross-entropy. During optimization, we alternated training between the cross-entropy objective and the hint loss at every batch, with virtual examples generated on the fly. Images from StyleGAN3 are generated at $512 \times 512$ resolution and then resized to $384 \times 384$ for training.

Table 7: Training and model hyperparameters for ViT-B/16 and Swin-B.

| Hyperparameter | ViT-B/16 | Swin-B |
|---|---|---|
| Resolution | $384 \times 384$ | $384 \times 384$ |
| Optimizer | AdamW | AdamW |
| Learning Rate | $1e{-}4$ | $1e{-}4$ |
| Weight Decay | 0.01 | 0.01 |
| Batch Size | 32 | 32 |
| Scheduler | Cosine Annealing | Cosine Annealing |
| Number of Epochs | 200 | 200 |
| Hint Loss Weight | 1.0 | 50.0 |

