# OpenReview forum: "Generative Hints"
_ICLR.cc/2026/Conference — Submitted to ICLR 2026_

### Official Review · Reviewer_HxYR · 2025-10-25

**Soundness:** 3
**Presentation:** 2
**Contribution:** 2
**Rating:** 6
**Confidence:** 3

**Summary:**

This paper introduces Generative Hints, a new training methodology that directly enforces known functional properties (called hints) in machine learning models using generative models. The key idea is to sample virtual examples from a generative model trained on the dataset and apply hint transformations (e.g., spatial or flip invariance) to them. These unlabeled virtual examples are used in a semi-supervised setup, combining the main classification loss with a hint loss that enforces the desired invariance (via symmetric KL divergence or MSE).

Unlike traditional data augmentation, which only indirectly encourages invariance through transformations on labeled data, generative hints explicitly regularize model behavior over the entire input space. The method reformulates a fully supervised task as semi-supervised learning without requiring unlabeled real data. Experiments across fine-grained visual classification datasets show the effectiveness of these methods.

**Strengths:**

1. The idea is original. Reinterpreting generative augmentation as functional constraint learning rather than sample diversity is innovative.

2. The formalism of hints, virtual examples, and invariance objectives is elegant and well presented.

3. The effectiveness of the method has been demonstrated across multiple architectures (transformers and CNNs) and tasks (vision, medical imaging).

4. Experimental studies include ablation on generative model quality and analysis of FID–correlation relationship.

**Weaknesses:**

1. Improvements over strong baselines (e.g., ViT/Swin with augmentation) are small (<2%), which may limit practical impact.

2. Results are constrained to medium-sized datasets; it remains unclear how this method performs on large-scale datasets (e.g., ImageNet).

3. The success of generative hints relies heavily on having a capable generative model. The cost and feasibility of training StyleGAN3 for every dataset is nontrivial.

4. There is no ablation on hint type: It would be helpful to understand how sensitive results are to the choice of hint function, weighting parameter, or temperature T.

**Questions:**

1. How does the method scale computationally when training high-quality generative models on large datasets? Could pretrained generative models (e.g., diffusion models) be reused effectively?

2. How sensitive are results to the choice of α and T? Would an adaptive scheduler materially change outcomes?

3. What happens when the generative model’s FID is worse than 15–20? Does the method ever degrade performance?

---

> ### Author Response · Authors · 2025-11-22
>
> Thank you for the thoughtful review and for recognizing the novelty of Generative Hints. Below we address each concern and question in detail.
>
> &nbsp;
>
> # W1) Modest Improvements (<2%)
>
> We agree the reported gains are modest. However, the key benefit of Generative Hints is that the improvements are **orthogonal and additive** to existing supervised training pipelines and do not require modifying the original data, architectures, or augmentation strategies. Since the hint loss operates exclusively on unlabeled generated images, the gains “stack” on top of strong baselines at effectively no cost to the supervised pipeline.
>
> &nbsp;
>
> # W2) Constrained to Medium-sized Datasets
>
> Our experiments focused on small and medium-scale datasets because these settings offer controlled environments for analyzing how properties are learned and how generative quality interacts with the hint mechanism. Most commonly-used image benchmarks fall into this category.
>
> While we have not yet scaled to ImageNet-sized datasets, we expect the method to extend naturally when strong pretrained diffusion models are available. Larger datasets often make the supervised task easier to learn the invariances, which may reduce headroom for improvement, but we believe the method remains applicable in principle.
>
> &nbsp;
>
> # W3) Reliance on training a Generative Model
>
> We acknowledge there is a prior in training a generative model per datasets introduces overhead. However, our results show improvements are achieved even with moderate generative quality (FID of 12.62 on Oxford Flowers, which has only 2,040 images and 102 classes).
>
> &nbsp;
>
> # W4)  Lack of Ablations on Hint Type, Weighting, and Temperature
>
> We agree that it is extremely important to introduce additional ablations on the hint. In the extended results included in the general response we added ablations across multiple hint types:
> - **Color Jitter Hint:** +0.76% on average (max +2.10%)
> - **Crop Hint:** +0.36% on average (max +1.20%)
>
> These results demonstrate that improvements are consistent across qualitatively different transformations and that the method is not narrowly used only for a specific hint function.
>
> &nbsp;
>
> # Q1) Computational Scaling and Use of Pretrained Generative Models
>
> Although we focused on StyleGAN3 for clarity and fast generation on medium-sized datasets, the method is compatible with pretrained generative models. When a pretrained model overlaps with the target distribution, we believe it can be used directly eliminating the training cost. However, we believe that this overlap in similarity distributions is important so that the invariance is learned with respect to the input distribution and not another arbitrary set of images.
>
> &nbsp;
>
> # Q2) Sensitivity to Alpha and Temperature Parameters
>
> In order to clarify the understanding of alpha and temperature we ran an ablation adjusting both of them for clarity as seen below. Due to the varying results depending on each of them this is where our intuition with an adaptive scheduler comes in. Furthermore, we believe the amount of emphasis will benefit greatly depending on where in the learning process it is and how much the hint has been learned.
>
> | Baseline / With Hints | α       | Accuracy (Alpha) | T     | Accuracy (Temp) |
> |------------------------|---------|------------------|-------|------------------|
> | Baseline (No Hints)    | --      | 91.11            | --    | 91.11            |
> | w/ Hints               | 1.0     | 91.52            | 0.5   | 92.05            |
> | w/ Hints               | 5.0     | 91.77            | 0.8   | 92.13            |
> | w/ Hints               | 10.0    | 91.85            | 1.0   | 91.80            |
> | w/ Hints               | 25.0    | 91.63            | 1.2   | 91.90            |
> | w/ Hints               | 50.0    | 92.13            | 1.5   | 91.58            |
> | w/ Hints               | 75.0    | 91.84            | 2.0   | 91.51            |
> | w/ Hints               | 100.0   | 92.02            | --    | --               |
>
> **Figure 1** *Performance sensitivity of generative hints when varying a and T independently, using the base hyperparameters from the original paper. While certain settings yield noticeably stronger gains, the overall trend shows consistent improvement across a broad range of a and T values, indicating robustness of the method rather than dependence on a narrow set of hyperparameters.*

---

> > ### Author Response · Authors · 2025-11-22
> >
> > # Q3) Behavior When Generative Model FID > 15-20
> >
> > We agree this is an important question. As shown in the extend ablation:
> > - Improvements are stable for FID  11.
> > - Moderate degradation occurs as FID enters the 12-15 range.
> > - Once FID exceeds ~15-20, the benefit diminishes and can eventually flatten or slightly regress.
> >
> > Crucially, the degradation is gradual, not catastrophic and the method never collapses the supervised performance. The hint loss becomes less informative, but it does not actively harm the classifier until the generative samples become extremely unrealistic.
> >
> > &nbsp;
> >
> > We appreciate your comments and have expanded our analysis to reflect these points. Looking forward to hearing your response.

---

> > ### Comment · Reviewer_HxYR · 2025-11-28
> > **Acknowledgment**
> >
> > Thank you for your response. I will keep my score.

---

### Official Review · Reviewer_G6Vz · 2025-10-27

**Soundness:** 2
**Presentation:** 4
**Contribution:** 2
**Rating:** 2
**Confidence:** 4

**Summary:**

This paper introduces Generative Hints, a novel approach to data augmentation. The authors train a generative model (StyleGAN3) and use its output, together with a transformation function h(x), to train a downstream classifier. The reported experiments demonstrate consistent improvements in classification accuracy across multiple datasets.

**Strengths:**

1. The paper proposes a new data augmentation method that leverages generative modeling.
2. The approach consistently improves classification performance across several benchmark datasets.

**Weaknesses:**

I find two major weaknesses in the current version of the paper. The first is that **the problem formulation contains significant ambiguity**, resulting in unclear or even contradictory statements throughout the paper. For example, in Section 3, Definition 3 fails to clearly define the generative model G. While I understand that the authors may refer to common generative models such as GANs or diffusion models, such assumptions should be explicitly stated for clarity and self-containment. Similarly, in Definition 4, the usage of the symbol “\approx” is never properly defined. Moreover, in Definition 7, if p and q represent probability density functions, then p/T is no longer a valid density function when T \neq 1, since \int p(x) dx / T \neq 1. These issues are not isolated but appear multiple times, making the overall formulation mathematically inconsistent and conceptually vague.

My second concern relates to the experimental section, which **lacks sufficient ablation studies to support the claimed performance improvements**. To be convinced that the proposed approach itself is responsible for the reported gains, I would expect to see experiments that vary hyperparameters, dataset scales, and the transformation function h, as well as analyses of failure cases. Without such analyses, it is difficult to attribute the improvement to the proposed idea rather than to specific settings or chance factors. Therefore, although the numerical results are encouraging, they are not yet convincing enough to demonstrate the effectiveness of the proposed method. I should also note that I am not an expert in image classification, so please disregard this comment if I have underestimated the technical contribution of the work.

_LLM Usage Disclosure:_
Note that this review was polished with assistance from a large language model (OpenAI GPT-5). The scientific assessment, judgments, and all substantive comments reflect my own independent evaluation.

**Questions:**

I have no further questions about this paper. Please address the weaknesses I mentioned above, especially those concerning the mathematical ambiguity and lack of ablation studies. Once these points are clarified, I will be happy to adjust the score accordingly.

---

> ### Author Response · Authors · 2025-11-22
>
> Thank you for taking time to go through and review our work. To your two main concerns:
>
> &nbsp;
>
> # W1) The problem formulation contains significant ambiguity
> Thank you for highlighting the ambiguities in our mathematical formulation. We have clarified the formulation and addressed each of your points below, providing updated definitions where relevant. We believe these refinements effectively resolve the potential ambiguities.
>
> &nbsp;
>
> ## W.1.1) Section 3, Definition 3 fails to clearly define the generative model G
> *Original:*
> **Definition 3 (Virtual Example).**
> A virtual example $x_v$ is an unlabeled input generated by a generative model $G$ trained on the training set $D_{\text{train}}$.
>
> *Revised:*
> **Definition 3 (Virtual Example).**
> Let $G$ denote a generative model (GAN, diffusion model, etc.) trained on the training dataset $D_{\text{train}}$.  A virtual example $x_v$ is any unlabeled image produced by sampling from $G$, i.e.,
> $$
> x_v = G(z), \quad z \sim p(z),
> $$
> where $p(z)$ is the latent prior of the generative model.   Thus, $x_v$ is a synthetic input that reflects the distribution learned from $D_{\text{train}}$ but does not have an associated ground-truth label.
>
> &nbsp;
>
> ## W.1.2) Definition 4, the usage of the symbol “\approx” is never properly defined
> *Original:*
> **Definition 4 (Invariance Hint on Virtual Examples).**
> Given a generative model $G$, an *invariance hint* is defined by a transformation function $h$ that maps a virtual example $x_v$ to a transformed example $x_v' = h(x_v)$.  The target function is assumed to satisfy the invariance property $f(x_v) \approx f(x_v'),$ for any virtual example $x_v$ sampled from $G$.
>
> *Revised:*
> **Definition 4 (Invariance Hint on Virtual Examples).**
> Given a generative model $G$ and a virtual example $x_v$ sampled from $G$, an invariance hint is defined by a transformation function $h$ that produces a transformed virtual example $x_v' = h(x_v)$. The target function $f$ is then encouraged to satisfy $ f(x_v) = f(x_v'),$ where the equality represents the constraint imposed in the learning objective, even though the true relationship between $f(x_v)$ and $f(x_v')$ may only hold approximately in practice.
>
> &nbsp;
>
> ## (W.1.3) Definition 7, if p and q represent probability density functions, then p/T is no longer a valid density function when T \neq 1, since \int p(x) dx / T \neq 1
> *Original:*
> **Definition 7 (Symmetric KL Hint Loss).**
> Let $h$ denote a hint transformation applied to a virtual example $x_v \in G$, producing $x_v' = h(x_v)$. Let the model's predicted probability distribution be  $\hat{f}(x_v) = p$ and $\hat{f}(x_v') = q$ where $\hat{f}$ is the model under training. The hint loss using symmetric KL divergence is defined as:
>
> $$
> \mathcal{L}_{\text{hint-ce}}(\tilde{p}, \tilde{q})
> = \frac{1}{2} \Big(
> \mathrm{KL}(\frac{p}{T} \| \frac{q}{T})
> +
> \mathrm{KL}(\frac{q}{T} \| \frac{p}{T})
> \Big).
> $$
>
>
> *Revised:*
> **Definition 7 (Symmetric KL Hint Loss).**
> Let $h$ denote a hint transformation applied to a virtual example $x_v \in G$, producing $x_v' = h(x_v)$. Let the model outputs  $\hat{f}(x_v) = p$ and $\hat{f}(x_v') = q$ denote the logits predicted by the model under training. The corresponding temperature-scaled probability distributions are defined as:
>
> $$
> \tilde{p} = \mathrm{softmax}\\left(\frac{p}{T}\right),
> \qquad
> \tilde{q} = \mathrm{softmax}\\left(\frac{q}{T}\right),
> $$
> where $T$ is a temperature parameter controlling the sharpness of the predictive distributions. The symmetric KL hint loss is:
> $$
> \mathcal{L}_{\text{hint-ce}}(\tilde{p}, \tilde{q})
> = \frac{1}{2} \Big(
> \mathrm{KL}(\tilde{p} \| \tilde{q})
> +
> \mathrm{KL}(\tilde{q} \| \tilde{p})
> \Big).
> $$
>
> &nbsp;

---

> > ### Author Response · Authors · 2025-11-22
> >
> > # (W2) Lacks sufficient ablation studies to support the claimed performance improvements.
> > We agree the ablation studies are important for isolating the contribution of the proposed method. In the general response, we provide a more complete set of results that address the points you highlighted. Here, we summarize the key finding:
> >
> > **1. Ablation across additional hint types.**
> >
> > Beyond the flip and spatial hint, we conducted ablations using a color-based hint and a crop hint. These yield consistent improvements of  on average +0.76% (up to +2.10%) and +0.38% (up to +1.20%) when using a color jitter and cropping hint respectively. This demonstrates that the method generalizes across qualitatively different transformations and is not tied to a particular choice of $h$.
> >
> > **2. Ablation on generative model quality.**
> >
> > We performed experiments varying the GAN fidelity to understand how performance scales with generator quality. Results show improvements are largely maintained when FID is below ~11, with a gradual performance drop, indicating that the method is robust until the generative model significantly degrades. The full results are included in the general response.
> >
> > &nbsp;
> >
> > Taken together, these support the claims that the improvements arise from Generative Hints rather than incidental factors. We are looking forward to your response and any additional questions you may have.

---

> > ### Comment · Reviewer_G6Vz · 2025-11-28
> >
> > Thank you for the response. I have reviewed the revised mathematical formulations. Although I appreciate the effort to improve the definitions, the current version still does not fully address my concerns, and I am not yet convinced that the score should be raised.
> >
> > While I understand the authors’ intended intuition behind these “definitions,” a mathematical definition must be stated without ambiguity. For instance, in the revision of Definition 3 (Virtual Example), you added “(GAN, diffusion model, etc.)”. The use of “etc.” is inappropriate in a formal mathematical definition because it leaves the scope undefined. Any mathematical concept requires a clearly specified form, range, or set; otherwise, the definition becomes imprecise. Such expressions may be acceptable in informal descriptions, but not in formal mathematical content.
> >
> > I therefore encourage the authors to reconsider whether these concepts truly need to be introduced as formal definitions. If they are mainly intended to offer intuition, a descriptive explanation, common in ML literature, may be more appropriate and clearer for readers.

---

> > > ### Author Response · Authors · 2025-12-01
> > >
> > > We acknowledge the reviewer’s point that the earlier wording of Definition 3 (Virtual Example), particularly the use of terms like “etc.”, is not appropriate for a formal mathematical definition. Because this definition is only meant to describe the setup of our generative-hints methodology rather than introduce a formal theoretical object, we will revise it to be a descriptive methodological explanation instead of a formal definition.
> > >
> > > This clarification does not affect the core contribution. Our updated experiments now include three distinct generative hints (spatial, color, crop), each showing consistent improvement over the data augmentation baseline. These results further support the robustness and generality of the method independent of the specific mathematical framing.
> > >
> > > In light of these adjustments and the strengthened empirical evidence, we hope the reviewer will reconsider the evaluation and adjust the score accordingly.

---

### Official Review · Reviewer_2eDL · 2025-11-01

**Soundness:** 2
**Presentation:** 4
**Contribution:** 2
**Rating:** 6
**Confidence:** 4

**Summary:**

This paper introduces Generative Hints, a semi-supervised training framework that leverages generative models to impose functional invariances on visual classifiers. The authors argue that conventional data augmentation only captures invariances locally within the training distribution, while a high-fidelity generative model can synthesize a broader set of plausible samples for enforcing invariance more effectively.

The proposed method trains a classifier using both (1) a supervised classification loss on labeled data and (2) a hint loss computed on generated, unlabeled images. The hint loss enforces consistency of model predictions under transformations corresponding to known invariances (e.g., horizontal flip, translation, small rotation). Empirical evaluations on fine-grained recognition benchmarks (CUB-200-2011, Stanford Cars, FGVC Aircraft, Oxford Flowers) and on CheXpert show consistent but moderate accuracy improvements over standard training with data augmentation.

**Strengths:**

1. Conceptually clear formulation of “hints.”
The paper formalizes the notion of a hint as a constraint linking inputs that should yield similar outputs, providing a unifying view that bridges data augmentation, regularization, and semi-supervised learning.

2. This  paper Novel use of generative models for invariance enforcement.
Rather than simply augmenting labeled samples, the approach explicitly uses a pretrained generator (StyleGAN3) to sample the data manifold and apply transformations in this synthetic domain, extending the effective training support.

**Weaknesses:**

1. Limited performance gains.
Reported improvements are relatively small (average ≈ +0.6 percentage points, maximum ≈ +1.8 points top-1 accuracy). While consistent, the benefits may not justify the additional computational overhead of training or maintaining a generative model.

2. Dependency on generator quality.
The approach relies heavily on the fidelity of the generator. The paper notes that when FID > 11, generative hints become ineffective, restricting applicability to domains with strong generative models.

**Questions:**

The paper formalizes hints as invariance constraints, but how does this differ in principle from standard consistency regularization used in semi-supervised learning (e.g., Mean Teacher, FixMatch)?

---

> ### Author Response · Authors · 2025-11-22
>
> Thank you for the positive review of our work. We address the noted weakness and questions below:
>
> &nbsp;
>
> # W1) Limited Performance Gain
>
> We agree the absolute improvements are modest (+0.6%, max +1.8%), but we want to emphasize:
> 1. **The gains are consistent across datasets and across multiple hint types.** In addition to the flip/spatial hint, we also obtained improvements of on average 0.76% (up to 2.10%) and 0.38% (up to 1.20%) using a color jitter hint and cropping hint respectively, demonstrating the effect generalizes beyond our initial hint.
> 2. **The benefits are “free” in that they are additive.** Generative Hints imposes no change to the supervised training pipeline. It operates exclusively on unlabeled generated images. Thus, they stack on top of existing augmentation or regularization techniques without interfering with them.
>
> &nbsp;
>
> # W2) Depending on Generator Quality
>
> We acknowledge the approach relies on reasonable generator quality. However, we are able to attain a strong performance with a FID of 12.62 on Oxford Flowers which has only 2,040 images with 102 classes showing there is potential benefit available. We include an ablation in the general response showing how performance scales with FID quality illustrating the degradation is gradual rather than catastrophic.
>
> &nbsp;
>
> # Q1) Differing from Standard Consistency in Semi-Supervised Learning
>
> Our method differs from classical semi-supervised consistency approaches (e.g., Mean Teacher, Fix Match) in two fundamental ways:
>
> 1. **Problem Setting.** We begin with a fully supervised problem, but introduce unlabeled generated examples, transforming the problem into a semi-supervised one.
> 2. **Nature of the constraint.** Our constraint is property-driven rather than augmentation driven. We explicitly enforce an image and its hint-transformed counterpart should yield approximately the same output distribution where the transformation is defined by a property of interest (e.g. flip invariance, spatial invariance, color stability).
> Mean Teacher and Fix Match enforces consistency under random augmentation and pseudo-labeling assumption, but they do not impose explicit property-based invariance constraints. In particular, they do not incorporate domain-specific invariance assumptions directly into the loss.
>
> &nbsp;
>
> Let us know any additional questions you would like us to clarify. Looking forward to your response.

---

### Official Review · Reviewer_Es7H · 2025-11-01

**Soundness:** 1
**Presentation:** 2
**Contribution:** 1
**Rating:** 0
**Confidence:** 4

**Summary:**

The idea of the paper is to present a generative data augmentation setup referred to as generative hints, which allows generative models to create extra images modeling the input distribution (but not being the same) and augmentations, that would allow capturing various properties of the target distribution.

In order to show the effectiveness of the idea, the paper utilizes StyleGAN3 to generate extra data that models the input distribution, and demonstrates two hint functions: Flip Invariance and Spatial Invariance (Translation and Rotation) -- modeling invariance properties between the input and target distributions. The invariance is imposed using a KL and a MSE loss.

Improved performance is demonstrated on the Stanford Cars, Caltech Birds (CUB), FGCV Aircraft and Oxford Flowers datasets, over the ViT-Base and Swin-Base models. Results are also shown on the CheXpert dataset using a ResNet50. In both cases, the baseline used is the standard version of the model; however, exact training specifics (aka data augmentation used on the training dataset, if any) are not provided.

**Strengths:**

The idea of creating "hints" or functions that capture diverse properties of the target distribution is strong and would help the community. However, the paper only presents invariance-based hints/transformations, which are already standard in the community.

**Weaknesses:**

The two main pillars of the presented paper are commonly existing ideas in the ML community, namely: generative (data) augmentation and data augmentation (for invariances). In my understanding, while the paper starts by planning to go beyond these ideas to present "other properties" (aka non-invariant properties like the authors specify for tabular data -- monotonicity), the proposed methodology and experiments do not go beyond existing knowledge to demonstrate any additional ideas or further analysis.

While the paper presents and claims a novel generative data augmentation paradigm, the experimental studies do not compare against any baselines beyond the authors' standard model training. Good baselines would include both existing generative [1,2,3,4] and non-generative data augmentation [5,6,7] techniques, as well as augmentations that impose transformation invariance [9, 10].

Furthermore, the paper claims lines 139-140 —"Applying hints directly on training data can lead to overfitting, where the model memorizes the hints with respect to specific training examples rather than learning the underlying property"—contradicts common knowledge of invariance-based input-data augmentation [9, 10], but fails to provide any evidence in support of this statement. I find the following additional baselines/ablations necessary for such a claim and good analysis of the reasons for improvement:
1. Input-data only invariance transformations
2. No invariance transformations, only generative data augmentation to the training set (class-based supervised loss)
3. No invariance transformations on input data; generative hints [proposed methodology]
4. Invariance transformations on all data (both input and generated)

The related work section misses out on numerous relevant works (not limited to the ones I mentioned) in the fields of:
1. Data Augmentation and Invariance: only two papers from 2017 and 2019 are cited, while the field has moved far ahead since. [5,6,7,8]
2. Generative Data Augmentation: only one paper from 2022 has been cited, while there has been immense progress in the field.[1,2,3,4]
3. Invariance transformations on the input data. [9,10,11].

[1] Bansal, H., & Grover, A. (2023). Leaving reality to imagination: Robust classification via generated datasets. arXiv preprint arXiv:2302.02503.

[2] Zheng, C., Wu, G., & Li, C. (2023). Toward understanding generative data augmentation. Advances in neural information processing systems, 36, 54046-54060.

[3] Azizi, S., Kornblith, S., Saharia, C., Norouzi, M., & Fleet, D. J. (2023). Synthetic data from diffusion models improves imagenet classification. arXiv preprint arXiv:2304.08466.

[4] Rahat, F., Hossain, M. S., Ahmed, M. R., Jha, S. K., & Ewetz, R. (2025, February). Data augmentation for image classification using generative ai. In 2025 IEEE/CVF Winter Conference on Applications of Computer Vision (WACV) (pp. 4173-4182). IEEE.

[5] Zhang, H., Cisse, M., Dauphin, Y. N., & Lopez-Paz, D. (2017). mixup: Beyond empirical risk minimization. arXiv preprint arXiv:1710.09412.

[6] Kim, J. H., Choo, W., Jeong, H., & Song, H. O. (2021). Co-mixup: Saliency guided joint mixup with supermodular diversity. arXiv preprint arXiv:2102.03065.

[7] Müller, S. G., & Hutter, F. (2021). Trivialaugment: Tuning-free yet state-of-the-art data augmentation. In Proceedings of the IEEE/CVF international conference on computer vision (pp. 774-782).

[8] Wang, L., Zhan, Y., Ma, L., Tao, D., Ding, L., & Gong, C. (2025). Splicemix: A cross-scale and semantic blending augmentation strategy for multi-label image classification. IEEE Transactions on Multimedia.

[9] Hounie, I., Chamon, L. F., & Ribeiro, A. (2023, July). Automatic data augmentation via invariance-constrained learning. In International Conference on Machine Learning (pp. 13410-13433). PMLR.

[10] Liu, Y., Yan, S., Leal-Taixé, L., Hays, J., & Ramanan, D. (2023). Soft augmentation for image classification. In Proceedings of the IEEE/CVF Conference on Computer Vision and Pattern Recognition (pp. 16241-16250).

[11] Quiroga, F., Ronchetti, F., Lanzarini, L., & Bariviera, A. F. (2018, January). Revisiting data augmentation for rotational invariance in convolutional neural networks. In International conference on modelling and simulation in management sciences (pp. 127-141). Cham: Springer International Publishing.

**Questions:**

Referring to my understanding of the paper (in summary), and why I think it's not novel and lacks good baseline comparisons (detailed in weaknesses). My ratings reflect this understanding of this work. If the authors can explain the novelty of their work beyond generative/non-generative data augmentation and invariance transformations (as in papers I've previously mentioned), I am open to a detailed discussion and to reevaluating my assessment of the paper.

There is also a lack of comparison with existing methods, no ablation study to back the claims in the paper (e.g., on which data subset should invariance or other transformations be applied), and the related work section is not thorough.

---

> ### Author Response · Authors · 2025-11-22
>
> Thank you for taking the time to review our work. We address all the concerns below as well as provide additional clarity into our generative hints methodology.
>
> &nbsp;
>
>
> # W1) Clarifying the Novelty beyond Standard Data Augmentation
>
> The core conceptual contribution of our paper is that Generative Hints are not a form of generative or non-generative data augmentation. Instead, they constitute a learning paradigm that uses unlabeled generative samples to explicitly enforce invariances through a hint loss, separate from the class-supervised loss.
>
> &nbsp;
>
> Unlike existing augmentation methods, our method does not only apply transformations directly to labeled training samples. Instead we:
> 1. **Sample unlabeled virtual examples** from a generative model *G*
> 2. **Apply a hint transformation** *h* to specify a property of interest (e.g. flip, spatial, color invariance)
> 3. **Enforce the property** through a symmetric KL hint loss (or MSE Loss)
> 4. **Switching off** between the supervised class loss on real data augmented labeled images and the hint loss on generated images.
>
> This formulation is orthogonal to existing augmentation methods, which modify the supervised training inputs. In contrast, our method uses unlabeled generative samples and a dedicated loss to explicitly encode the desired property.
>
> Thus, Generative Hints do not replace augmentation. They complement it and can be added on top of any existing augmentation pipeline.
>
> &nbsp;
>
>
> # W2) Reviewers Request for Baselines
>
> We want to further clarify the reasoning for our use of baselines. Based on our previous problem formulation, since we are orthogonal to existing methods we believe the proper baseline is:
> 1. **(Data Augmentation Baseline)** Invariance transforms training only on the input data.
> 2. **(Generative Hints)** Same invariance transforms training on both the transformed input data (as in 1) as well as doing invariance training on generated data using the same invariance transformation, but training for the consistency.
>
> Additionally, generative data augmentation acts as a separate class of transformation and should be viewed as complementary to generative hints, rather than a direct point of comparison.
>
> Our method was able to get consistent improvements of on average +0.63% (up to +1.78%), 0.76% (up to +2.10%), and +0.38% (up to +1.20%) over standard data augmentation with a spatial hint, color jitter hint, and crop hint respectively. We point to the general response for the full set of results.
>
> &nbsp;
>
> # W3) Clarifying our Comment on Overfitting
> Referring to the point “Applying hints directly on training data can lead to overfitting . . .”:
>
> The reviewer noted a statement regarding overfitting. Our comment referred specifically to the semi-supervised scenario where applying the hint loss directly on labeled training inputs could cause the model to memorize the exact outputs rather than learn the underlying property. This is different from the supervised augmentation approaches discussion in prior work. We will update this statement to address this clarity.
>
> &nbsp;
>
> # W4) Concern Around Related Work
>
> We recognize that our related work section did not fully cover recent advances in modern augmentation and generative data augmentation methods. Our focus was to present an alternative to data augmentation rather than an improvement upon it, and our approach is largely agnostic to the specific generative method used due to its robustness to label quality. We will expand the related work section to ensure completeness based on your suggestions.
>
> &nbsp;
>
> We believe we address the reviewer’s concerns and are looking forward to your response.

---

> > ### Comment · Reviewer_Es7H · 2025-11-26
> > **Concerns still unresolved**
> >
> > ## Novelty
> > I reiterate my initial concerns below, which I do not believe have been addressed. No empirical or theoretical evidence has been provided that counters them.
> >
> > The idea of *generative hints* imposing a function over the input domain is interesting. However, the theoretical formulation is very simple and already well-known: **virtual examples** (i.e., generated samples) and **invariance-based losses** (i.e., contrastive learning).
> >
> > While the formulation of generative hints *could* encapsulate a broader range of functions, the paper empirically demonstrates **only invariance-based functions/hints**. Injecting biases into vision models is a long-standing goal in the community, and since the paper does not go beyond demonstrating the effectiveness of well-known invariance-based losses, **the novelty remains unclear**.
> >
> > Given this, (supervised) co-training with generative hints is **equivalent to contrastive learning on generated samples**, a topic with extensive prior literature—which I cited in the initial review.
> >
> > --
> >
> > ## Orthogonality and Baselines
> >
> > As cited in my original review, several prior works directly address **using generative samples to improve classification performance**, which is exactly the task in this paper. Given the substantial overlap in both approach and objective, these works are **necessary baselines**.
> >
> > Even if the authors claim orthogonality, it must be **demonstrated**.
> > To claim orthogonality, empirical evidence must show that the proposed methodology delivers improvements **beyond prior work** when combined with one. This has not been shown in either the paper or the authors’ response.
> >
> > --
> >
> > ## Lack of Requested Ablations
> >
> > In my original review, I requested experiments and ablations that would support the paper’s claims. These have not been provided.
> >
> >
> > Quoting authors response to my overfitting query: _"applying the hint loss directly on labeled training inputs could cause the model to memorize the exact outputs rather than learn the underlying property."_
> > This needs to supported by empirical evidence, or removed as a claim. In fact I had requested this exact ablation to be provided in my review, I will reiterate them below.
> >
> > I listed the following **necessary baselines/ablations**:
> >
> > 1. Input-data-only invariance transformations
> > 2. No invariance transformations; only generative data augmentation (class-based supervised loss)
> > 3. No invariance transformations on input data; generative hints (the proposed method)
> > 4. Invariance transformations on all data (input + generated)
> >
> > I appreciate the additional results on color jitter and cropping. However, these are *still standard invariance-based transformations*, widely used in contrastive learning and data-augmentation. A **non-invariance-based transformation** must be demonstrated empirically to claim contributions beyond existing knowledge.
> >
> > --
> >
> > ## Clarification on the Training Paradigm
> >
> > The training details in Section 5.2 and Appendix A.3 do not specify how long the models (with and without hints) were trained (epochs or training samples).
> >
> > - Did both models see the **same** number of training samples (the one with hints may have seen more overall)?
> > - Why were ImageNet-1k pretrained models used?
> > - Do hints still help when models are trained from scratch?
> >
> > Given the empirical evidence, the proposed method is essentially **supervised learning + contrastive learning on generated samples**, which has been demonstrated before. The paper also includes a very limited experimental evaluation, lacks key baselines, and provides no empirical support for the orthogonality claim.

---

> ### Author Response · Authors · 2025-12-01
>
> # Novelty
> The papers cited by the reviewer fall into established augmentation categories:
>  - **Generative data augmentation** using GANs or diffusion models [1–4],
>  - **Mixup-style** data augmentation methods [5–8],
>  - **Learned Data Augmentation** invariance-based approaches for class supervision [9–11].
>
> None of these works involve any form of contrastive objective, generative or otherwise. In all cases, generated or transformed samples are incorporated directly into the **supervised class loss**, changing the effective training distribution rather than imposing an explicit functional constraint.
>
> &nbsp;
>
> Our method differs in two fundamental ways:
>
> 1. **Independent functional loss vs. modifying the supervised objective:** Existing generative augmentation methods modify the supervised loss by inserting generated samples directly into it. In contrast, we do not alter the supervised class loss at all. The hint loss is a fully independent objective applied only to generated samples, allowing it to be added on top of any supervised pipeline without interference.
>
> 2. **Contrastive Learning Difference, Prediction-space vs. feature-space:** Contrastive learning methods, whether using real or generated views, operate in **feature space**, enforcing that the embeddings of related samples are similar. The objective is to shape the representation geometry through similarity constraints such as InfoNCE or latent alignment. In contrast, **generative hints operate in prediction space**. Rather than aligning embeddings, generative hints require the classifier to satisfy a functional constraint of the form $f(x)=f(h(x))$ for a hint transformation $h$. This enforces task-level behavior directly at the output level, not representation-level similarity. As a result, generative hints influence the decision function, not the embedding structure, making them fundamentally different from contrastive learning approaches.
>
> &nbsp;
>
> Importantly, **we have found no prior work that uses unlabeled generated examples to enforce an additional loss supervised learning setting**, without altering the supervised loss or the training distribution. This separation, using generated data as explicit functional hints rather than as augmented training inputs, is fully novel.
>
> &nbsp;
>
> # Orthogonality, Baselines, and Requested Ablations
> The cited augmentation papers apply generated or transformed samples directly within the supervised class loss. In contrast, our method does not modify, replace, or interact with the supervised loss at all. Generative hints introduce an entirely separate prediction-space loss applied only to generated samples, enforcing functional consistency independently of any augmentation strategy used on real data. This clean separation is what makes the method orthogonal: it can be added on top of any supervised training pipeline without altering it.
>
> &nbsp;
>
> Because of this design, the reviewer’s requested baselines—such as applying invariances directly to input data or replacing augmentations—do not correspond to our methodology. Our comparison is exactly the one that reflects the method’s structure: **baseline pipeline (with any standard augmentations) vs. baseline + generative hints**, holding the supervised component fixed. Generative hints are not an alternative to augmentation; they are an additional functional constraint that complements whatever supervised pipeline is already in place.
>
> &nbsp;
>
> Finally, although the formulation supports non-invariance hints, our revision provides three distinct invariance hints (color, crop, rotation), each showing consistent improvement, demonstrating that the mechanism is robust.
>
> &nbsp;
>
> # Clarification of the Training Regime
> We already provided training specifications (training/test split in Table 4; learning details in Table 7), and all models, baseline and generative-hints variants, are trained for 200 epochs. Both methods see the same number of real training samples. The hints model additionally applies the hint loss on generated samples, but this does not affect the supervised pipeline.
>
> &nbsp;
>
> Using ImageNet-1k pretrained backbones is standard practice for small and medium-sized vision datasets. Since generative hints introduce functional regularization, they are expected to maintain or improve performance even when training from scratch.
>
> &nbsp;
>
> We believe this clarification directly addresses the reviewer’s misunderstanding of our method and resolves their concerns.

---

### Author Response · Authors · 2025-11-22
**Official Rebutal**

We thank all the reviewers for their thoughtful feedback and constructive suggestions. Below we summarize the core contributions of our work and highlight the additional ablation studies we performed in response to reviewer comments.

&nbsp;

# Summary of Contributions

Our goal is to learn an image classification that has known invariance properties to transformations such as flips, rotations, spatial shifts, color jitter and cropping. Traditional approaches enforce these properties indirectly through supervised data augmentation on labeled training images. Recent generated generative augmentation methods still rely on adding transformed labeled samples to the training pipeline.

&nbsp;

**Generative Hints introduce a different paradigm.**

Instead of transforming labeled training data, our method follows:
1. **Samples unlabeled virtual examples** form an image generative model $G$ (GAN, Diffusion)
2. **Applies a hint transformation** *h* that encodes a desired property
3. **Enforces the property explicitly** via a symmetric KL (or MSE) hint loss
$ f(x_v) = f(h(x_v)), \quad\text{where } f \text{ is the target function and } x_v \text{ is the virtual example.} $
4. **Alternates** between the supervised class loss on real images and the hint loss on generated images.

This creates a *semi-supervised* setup where generative samples carry property-based constraints, making the method orthogonal and additive to existing augmentation or regularization techniques.

&nbsp;


# Ablation Across Hint Types

To demonstrate that Generative Hints are not specific to a spatial/flip invariance hint, we added two new hint types:
- **Color Jitter Hint:** Random brightness/contrast/saturation adjustment up to ± 20%.
- **Crop Hint:** Resize to 448x448 and random crop to 384x384.

Using the same setup as our main result (ViT-B and Swin-B on Stanford Cars, FGVC Aircraft, Caltech Birds, and Oxford Flowers), we observe for the color jitter hint and crop hint improvement on average of +0.76% (up to +2.10%) and +0.38% (up to +1.20%) respectively. The full results using the color jitter hint and crop hint can be seen in figures 1 and 2 below.

&nbsp;

| Dataset         | ViT-B Baseline | ViT-B w/ Hints | Swin-B Baseline | Swin-B w/ Hints |
|-----------------|----------------|----------------|------------------|------------------|
| Stanford Cars   | 89.15          | **90.29**      | 91.11            | **92.13**        |
| FGVC Aircraft   | 82.51          | **84.61**      | 90.23            | **90.66**        |
| Caltech Birds   | 87.92          | **88.37**      | 90.06            | **90.58**        |
| Oxford Flowers  | 99.14          | **99.45**      | 99.58            | **99.66**        |

**Figure 1.** *Results using the Color Jitter hint on Vit-B and Swin-B across Stanford Cars, FGVC Aircraft, Caltech Birds, and Oxford Flowers. The method yields consistent improvements, averaging +0.76% (up to +2.10%).*

&nbsp;


| Dataset         | ViT-B Baseline | ViT-B w/ Hints | Swin-B Baseline | Swin-B w/ Hints |
|-----------------|----------------|----------------|------------------|------------------|
| Stanford Cars   | 90.39          | **90.43**      | 92.87            | **93.18**        |
| FGVC Aircraft   | 82.37          | **82.44**      | 90.74            | **91.11**        |
| Caltech Birds   | 87.91          | **89.11**      | 90.57            | **90.96**        |
| Oxford Flowers  | 98.89          | **99.50**      | 99.62            | **99.67**        |

**Figure 2.** *Results using the Crop hint on Vit-B and Swin-B across Stanford Cars, FGVC Aircraft, Caltech Birds, and Oxford Flowers. The method yields consistent improvements, averaging +0.38% (up to +1.20%).*

&nbsp;

# Scaling Behavior with Generative Model Quality (FID)

Several reviews asked how performance scales as the generative model’s FID worsens. We performed an ablation using Swin-B with the Color Jitter Hint on Stanford Cars while varying GAN quality. The results show that the stable and beneficial performance when FID  12 and gradual degradation as FID increases and a steep drop off once FID > 15. The full results can be seen in Figure 3 below.

| Setting             | FID ↓ | Accuracy (%) ↑ |
|---------------------|-------|----------------|
| Baseline (No Hints) |  --   | 91.11          |
| w/ Hints            | 5.29  | 92.13          |
| w/ Hints            | 11.68 | 91.83          |
| w/ Hints            | 19.15 | 91.33          |
| w/ Hints            | 30.83 | 91.09          |

**Figure 3.** *Effect of generator quality (FID) on the final performance using the Color Jitter Hint with Swin-B on Stanford Cars. Performance remains stable and beneficial at moderate FID levels before declining as generative quality degrades.*

&nbsp;

We thank the reviews again for their feedback and believe the additional analyses strengthen both the clarity and the empirical support of our work.

---

### Meta-Review · Area_Chair_etjT · 2026-01-09

**Summary:**

This paper proposes Generative Hints, a training procedure that uses samples from a dataset-trained generator as unlabeled “virtual examples” and applies a consistency-style “hint” loss in prediction space to enforce known invariances. Reviewers agree the framing is clear and the empirical improvements are generally consistent but modest; however, there are unresolved concerns regarding novelty and convincing experimental evidence. In particular, Es7H maintains that the method is substantially overlapping with well-established consistency/contrastive-style invariance enforcement using synthetic views, and argues that the paper does not empirically demonstrate either the claimed “orthogonality” to existing generative augmentation/consistency methods or advantages beyond standard invariance losses; key requested baselines/ablations are still missing. A second reviewer (G6Vz) highlights continuing issues with the formal/mathematical presentation and remains unconvinced despite revisions, suggesting the “definition-heavy” exposition is not yet precise or appropriate. While the authors add helpful ablations and address some presentation issues, these additions do not resolve the core attribution problem: it remains unclear whether the reported gains are specifically due to the proposed paradigm versus known consistency regularization on additional samples, and the evaluation lacks the comparative baselines needed for a strong claim of novelty and impact. Given the mixed but overall low scores and the persistence of these central concerns after rebuttal, I recommend rejection, encouraging resubmission with (i) stronger, up-to-date baseline comparisons, (ii) the ablations needed to isolate the contribution and validate “orthogonality,” and (iii) a streamlined, unambiguous methodological presentation.

**Reviewer Concerns:**

The authors added some ablations and clarified training length/setting to a certain extent. They also partially addressed math issues; agreed to drop overly formal “definitions” and reframe descriptively. Yet, the rebuttal lacks a convincing argument for novelty especially with respect to prior consistency/contrastive-style invariance learning strategies; furthermore, the “orthogonality” claim is not empirically demonstrated; key requested baselines/ablations vs generative augmentation and modern augmentation methods still missing. Last, formal precision concerns persist (still not fully satisfied), and practical impact remains limited.

**Reviewer Scores:**

In my assessment of the review + rebuttal period, it is very unlikely that the negative scores would have changed much in case of a full discussion. One reviewer with a "Marginally above ..." score even explicitly mentions that the score remains unchanged.

---

### Decision · Program_Chairs · 2026-01-26

Reject